# Mixed Models in Nonlinear Regression for Description of the Growth of Nelore Cattle

**DOI:** 10.3390/ani13010101

**Published:** 2022-12-27

**Authors:** Raimundo Nonato Colares Camargo Júnior, Cláudio Vieira de Araújo, Welligton Conceição da Silva, Simone Inoe de Araújo, Raysildo Barbosa Lôbo, Lílian Roberta Matimoto Nakabashi, Letícia Mendes de Castro, Flávio Luiz Menezes, André Guimarães Maciel e Silva, Lílian Kátia Ximenes Silva, Jamile Andréa Rodrigues da Silva, Antônio Vinicius Correa Barbosa, José Ribamar Felipe Marques, José de Brito Lourenço Júnior

**Affiliations:** 1Postgraduate Program in Animal Science (PPGCAN), Institute of Veterinary Medicine, Federal University of Para (UFPA), Federal Rural University of the Amazon (UFRA), Brazilian Agricultural Research Corporation (EMBRAPA), Castanhal 68746-360, PA, Brazil; 2Department of Agricultural and Environmental Sciences, Federal University of Mato Grosso (UFMT), Sinop 78550-728, MT, Brazil; 3National Association of Breeders and Researchers, Ribeirão Preto 14020-230, SP, Brazil; 4Department of Veterinary Medicine, Federal University of Para (UFPA), Castanhal 68746-360, PA, Brazil; 5Institute of Animal Health and Production, Federal Rural University of the Amazônia (UFRA), Belém 66077-830, Brazil; 6Brazilian Agricultural Research Corporation (EMBRAPA), Belém 70770-901, PA, Brazil

**Keywords:** beef cattle, Brody, longitudinal data

## Abstract

**Simple Summary:**

Mixed nonlinear models are an extension of mixed models combined with nonlinear models, allowing one or more parameters of the nonlinear regression function to be considered both as fixed and random effects. This work was carried out with the objective of comparing the use of several nonlinear functions of the fixed and mixed linear models to determine the body growth of both sexes in Nelore cattle. For this, body weight records were used to describe the growth curve of the animals. The Brody function obtained the best adjustments according to the evaluators, finding a reduction in the residual variance of 79 and 83%, for males and females, respectively. The absolute growth rate for males ranged from 0.921 to 0.261 kg/day and for females from 0.922 to 0.198 kg/day, while the relative growth rate ranged from 2.55 to 0.06% for females and from 2.39 to 0.08% for males. Males showed greater growth acceleration at the beginning of the growth trajectory, equaling the females at 397 days of age, and thereafter presented lower estimates. Nonlinear regression in the context of mixed models allows for a reduction in residual variance and an increase in model precision.

**Abstract:**

Body weight records were used to characterize the growth curve of Nelore cattle. Body weight was regressed as a function of age, for both sexes, by using nonlinear models through the functions of Brody, Gompertz, Logistic, Richards, Meloun 1, Von Bertalanffy, and Von Bertalanffy. The quality of the model arrangements was evaluated by employing Akaike and Bayesian Schwarz information criteria. The Brody function provided the best adaptations by the evaluators and, considering the asymptotic weight and the maturation rate as random, a reduction in residual variance of 79% for males and 83% for females was obtained in relation to the models under fixed contexts. In males, the absolute and relative growth rates ranged from 0.921 to 0.261 kg/day and 2.39 to 0.08%, respectively. For the same rates, under another approach, females ranged from 0.922 to 0.198 kg/day and 2.55 to 0.06%, respectively. Males showed greater growth acceleration at the beginning of the growth trajectory, being equal to females at 397 days of age and from that age onward they presented lower estimates. The nonlinear regression model approach under the mixed-models context allows reduction of residual variance, increasing model accuracy.

## 1. Introduction

Success in beef cattle production systems is determined by a strategic and efficient control of production, such as animal weight development, because all production processes are directly related to weight development. Usually, this description is made through growth curves in which the body weight of the animal is related to its age [1,2].

Animal body growth is a relevant event in beef cattle, as its knowledge favors the management of information regarding the different formats of animal growth curves, allowing the identification of animals with earlier growth [3].

The analysis of repeated measures data (longitudinal data) regressing the body weight of animals as a function of age through nonlinear models, in the description of growth curves, has the benefit of summarizing all the known data on weight performance from individuals to a group of few parameters that can be biologically explained and, consecutively, can be available to derive other relevant characteristics of animal growth, for example, the absolute growth rate [4].

The use of mixed nonlinear models in the description of the growth curve of the animals is an interesting methodology, because the incorporation of random effects, in addition to the error, can considerably improve the model’s adjustment by reducing the residual variance [5].

Mixed nonlinear models are an extension of mixed models combined with nonlinear models, allowing one or more parameters of the nonlinear regression function to simultaneously be considered as both fixed and random effects. As a result of all that was exposed, this work was carried out with the objective of comparing the use of several nonlinear functions of the fixed and mixed linear models to determine the body growth of both sexes in Nelore cattle.

## 2. Materials and Methods

### 2.1. Animal Registration

A total of 37,416 records of body weight at birth, 120, 210, 365, 450, and 550 days of age in animals of the Nelore breed were used, with 17,328 records of male animals and 20,088 records of female animals, from herds raised in the Midwest Region of Brazil, participants in the Programa de melhoramento genético (genetic improvement program) da Associação Nacional de Criadores e Pesquisadores–ANCP (National Association of Breeders and Researchers).

### 2.2. Body Weight

Animals’ body weight was regressed by their age (in days) and separately by gender, by using the modified nonlinear functions of Brody, Gompertz, Logistic, Richards, Meloun 1, Von Bertalanffy and Von Bertalanffy (Table 1).

Asymptotic weight (A) can be thought of as the mean weight at maturity. The parameter B is an integration constant that, as a rule, has no biological interpretation in most functions. The parameter K is interpreted as the rate of maturation, which can be defined as the change in weight in relation to weight, and maturity can be understood as an indicator of the growth rate with which the animal reaches its adult size.

### 2.3. Quality of Adjustment Evaluators of Nonlinear Models

The quality of adjustment evaluators of the nonlinear models used were (1) Akaike information criterion (AIC), AIC= −log⁡(L) + 2p, where log⁡(L) is the logarithm of the likelihood function of the density function of probability. (2) The Schwarz Bayesian information criterion (BIC) is expressed as BIC = −2log⁡(L) + p log⁡〖(n)〗.

### 2.4. Statistical Analysis

Initially, all models were processed considering the function parameters as fixed, after choosing the function that best described the growth curve of male and female body weight; it was used in the context of mixed models, considering the possible parameters of biological interpretation (parameters A and K), individually or together as random.

Under the context of mixed models and considering y_ij_, the j-th measurement of body weight in the i-th individual and t_ij_ the age (in days). The regression model presents residuals in a normal distribution with zero mean and constant variance $\sigma_{e}^{2}$. Parameters A and K were considered as random with normal distribution and B a fixed parameter.

The measures y_ij_ are independent in relation to the index i, but not in relation to j, given that fixing i, the measures y_ij_ are taken longitudinally for a single individual. Thus, it becomes necessary to incorporate intraindividual variance components into the model.

Considering that A_i_~N (A; $\sigma_{a}^{2}$) and K_i_~N (K; $\sigma_{K}^{2}$), where $\sigma_{a}^{2}$
e $\sigma_{K}^{2}$ are variance components of parameters A and K. In addition, M = (A_i_; K_i_), a random-effect vector and *N′* = (A, B, K; $\sigma_{e}^{2}$), a fixed effect vector; *f(y_i_|t_i_,N,M_i_)ω(M_i_;Σ*) is the joint probability density function, where $y_{i}^{\prime }$ = (*y_i_*_1_*, y_i_*_2_*, …, y_ij_*), $t_{i}^{\prime }$ = (*t_i_*_1_*, t_i_*_2_*, …, t_ij_*), *∑′*
*=* ($\sigma_{a}^{2}$, $\sigma_{K}^{2}$), and *_Ɯ_* is the joint density of *Ai* and *Ki.* The marginal likelihood function is given by $L\left(N;\Sigma \right)=\prod_{I-1}^{n}\;\int f(y_{i}|t_{i},N,M_{i})\varpi \;\left(M;\sum \right)dM_{i}$.

Estimators for the parameters in N and *∑* are obtained by maximizing *L*(*N;∑*) in relation to these quantities. The procedure PROC NLMIXED of software SAS [6] numerically minimizes −*L*(*N;∑*) in relation to the parameters N and *∑* being the variance matrix and approximate covariances for the estimators obtained by the inverse of the Hessian matrix.

For joint analysis, considering simultaneously the two parameters, A and K as random in the model, the assumptions about these parameters are described as $\left[\begin{array}{c} A_{i}\\ K_{i} \end{array}\right]\;\sim NM\left(\left[\begin{array}{c} A\\ K \end{array}\right],\left[\begin{array}{cc} \sigma_{a}^{2} & \sigma_{a,b}^{}\\ \sigma_{b,a}^{} & \sigma_{b}^{2} \end{array}\right]\;\right)$, in which $\sigma_{a,b}^{}$ = $\sigma_{b,a}^{}$ is the covariance between parameters A and K.

All analyses were performed by using the Nlin and Nlimixed procedures of the SAS software [6].

## 3. Results and Discussion

Estimates of averages, standard deviations, and coefficients of variation were calculated for body weight at birth—120, 210, 365, 450, and 550 days of age, by gender (Table 2). The largest differences observed for body weight between genders occurred after 120 days of age.

Males became heavier and more heterogeneous than females with aging. This fact is explained due to homeorhesis that sets out the regulation of new priorities for the animal’s body, such as the distribution of nutrients from the mother’s body to fetal tissues or to milk production, thus altering the growth curve [7,8].

Estimates of function parameters that describe the growth curve of male and female body development for each function (Table 3).

The average weights at 550 days of age were 332.63 ± 41.15 and 301.76 ± 32.63 kg, for males and females, respectively. Considering that the age range studied did not include animals in adulthood, it can be said that the Brody and Meloun I model, which presented the same asymptotic weight, presented estimates closer to the adult weight in the Nelore breed of 514.24 ± 59.25 kg, as estimated by Boligon et al. [9]. On the other hand, in the logistic function, this parameter was underestimated in both genders. A similar value of asymptotic weight, also in the Nelore breed, was obtained by Lopes et al. [10].

In all functions, females had lower asymptotic weights, but with higher maturity rates represented by the K parameter. The K parameter is interpreted as an indicator of the speed with which the animal reaches its adult size; that is, an indicator of the precocity of the animals. McManus et. al. [11] argues that the most important biological relationship for a curve is between parameters A and K.

Correlations between asymptotic weight (A) and maturation rate (K) were negative for all nonlinear functions, ranging from −0.84 and −0.76 (logistic) to −0.98 and −0.97 (Brody) in males and females, respectively.

The antagonistic relationship between asymptotic weight and maturity rate indicates that animals with greater precocity of growth reach lower body weights at maturity, as between females and males, females were earlier, but with lower adult weight compared to males. The same antagonistic behavior was verified by Malhado et al. [12] and Lopes et al. [10]. Souza et al. [13] found an estimated correlation between parameters A and K equal to −0.62, indicating that earlier animals are less likely to reach high weights in adulthood.

On the other hand, Silva et al. [14] found higher estimates of asymptotic weight (498.97) and maturation rate (0.0494) for Nelore females with body weight measured from birth to 100 days of age. The expected genetic correlation between asymptotic weight (A) and maturity rate (k) should be negative, as heavier animals tend to have a lower maturation rate than lighter animals, which, in turn, must reach maturity in younger ages [15].

According to the criteria adopted as evaluators of the adjustment quality of the functions in the description of the growth curve of the body weight of the animals, the estimates of Asymptotic Index (AI), adjusted coefficient of determination (R²_ij_), Akaike Criterion (AIC) and Schwarz’s Bayseian Information Criterion (BIC), indicated that the functions of Brody and Meloun I are the ones that presented the best values of estimates of the criteria. Both functions identically estimated the asymptotic weight and the maturation rate, as well as the same values for all evaluation criteria of the best function (Table 4).

The Brody function was chosen as the most adequate to explain the body weight growth curve for both sexes. This is because, in addition to being the most used in the description of growth curves in animals, it is still very similar to the Meloun I function, because only the constant parameter of integration differentiates them. At the other extreme, also for both sexes studied, the logistic function presented the worst estimates for the evaluation criteria of the goodness of fit of the functions.

However, according to the adjusted coefficient, the determination estimates—all greater than 0.90 despite the differences in asymptotic weight and estimated maturation rate by the different functions—all describe the body weight growth curve in the studied interval in a very similar way.

When using different models (Brody, Von Bertalanffy, Logistic, and Gompertz) to describe the growth curve of Nelore cattle, it was observed that they all converged, although Brody’s had the best fit [14].

Souza et al. [13] adjusted growth curves for animals of the Indubrasil cattle breed and found a better adjustment for the logistic model, followed by the Gompertz and Von Bertalanffy models.

Arango and Van Vleck [17] stated that the Brody function, which despite being less sensitive to fluctuations in weight, is more suitable for modeling the growth curve of cattle, because the results are easier to obtain and interpret. On the other hand, Lopes et al. [10] and Santana [18], using the functions of Brody, Gompertz, Logistic and Von Bertalanffy to describe the growth curve of Nelore animals, found better adjustment through the model that used the von Bertalanffy function. Still, Mazzini et al. [19], when analyzing data on weights and ages of animals of the breed of European origin, found that the Brody function underestimated the average weight of the animals.

There was a reduction in the estimate of asymptotic weight and increased maturation rate in both genders (Table 5) when regressing the body weight of the animals as a function of age and separately by gender. We used the Brody function, however, considering the asymptotic weight and maturation rate (A and K, respectively) as random. However, for the age range studied, the description of the body weight growth curves of males and females were very similar in the approach of fixed and mixed models.

The criteria for assessing the adjustment quality of the AIC and BIC functions, due to the reduction of the estimate of the natural logarithm of the likelihood function of the probability density function, pointed to nonlinear regression models using the Brody function in the context of mixed models, considering asymptotic weight and maturation rate as random, more suitable for describing the growth curve of male and female body weight.

Consequently, the approach of mixed models in nonlinear regression that used the Brody function led to the partition of the total variance into more components as well as the error, causing a reduction in the estimates of residual variances by approximately 79 and 83% for males and females, respectively, when compared to the same estimates obtained in the context of fixed models.

Applying the methodology of mixed nonlinear models to adjust the growth curve of dairy goats, Moreira et al. [20] verified that the coupling of parameters with biological meanings to random effects in the functions of Brody, Van Bertalanffy and Richards, reduced the residual variance, making the estimates more credible, similarly to what was obtained in this study.

Differences in body weight predicted by the mixed model of males compared to females have been expressed as a percentage (Figure 1). The greater difference between body weights observed at birth (males 6.1% heavier than females) is because many ranchers record predetermined weights for males and females at birth and not the actual weights. It is observed that during the lactation period, the differences in body weight between males and females are smaller when compared to the differences observed with aging, and at 550 days of age, males were 9.35% heavier than males in relation to females.

The absolute growth rates (AGR) obtained by the first derivative of the Brody function, under the context of mixed models, in relation to the age of the animals (in days) for males $\;(AGR=\left(434.4\right)\left(0.9242\right)\left(0.0023\right)e^{-0.0023x_{i}}$) and for females ($AGR=\left(362.13\right)\left(0.9123\right)\left(0.0028\right)e^{-0.0028x_{i}}$). in which x_i_ is the age of the animals in days, are shown below (Figure 2).

The absolute growth rate reflects the increase in body weight from birth to the point at which growth is maximal, which corresponds to the tipping point. Subsequently, it decreases until reaching values close to zero when the maximum size of the individual is reached (asymptotic weight).

As in the Brody function the inflection point coincides with the birth, that is, it is a derivation of the Richards function, in which the parameter that shapes the curve (m), in the Brody model m is equal to 1, resulting in a non-sigmoid curve. Thus, this model describes, in both genders, a phase of self-deceleration of growth that occurs after birth (inflection point).

AGR corresponds to the average weight gain of the animals, obtained per unit of time estimated along the growth trajectory and indicates that males and females showed similar daily gains at the beginning of the growth trajectory, with 0.921 and 0.922 kg/day, respectively. With aging, males showed greater daily gains compared to females, 0.261 and 0.198 kg/day, respectively.

The result confirms again that females with greater precocity of weight gain (higher estimate of maturation rate) will have lighter carcasses at the point of slaughter (lower AGR).

Lopes et al. [10], using the Brody model, found AGR equal to 0.753 kg/day and 0.235 kg/day, respectively, at the beginning and end of the body growth trajectory of Nelore animals.

The relative growth rate (RGR), obtained as the ratio between the AGR and the body weight predicted by the nonlinear function, is expressed as the proportion of the increase in the animal’s weight for each day in relation to its predicted weight, that is, how much the animal gained in body weight at age x_i_, in relation to its existing body weight at that age (Figure 3).

Females showed higher RGR compared to males at the beginning of the growth trajectory (up to approximately 30 days) with RGR of 0.0255 and 0.0239 for females and males, respectively, for ages equal to 1 day. That is, the mass gain was proportional to 2.55% and 2.39%, for females and males, respectively, in relation to the body mass they already had.

The RGR values decreased with aging, with estimates equal to 0.06 and 0.08% for females and males, respectively, at 550 days of age. A similar behavior of RGR was observed by Lopes et al. [10].

The growth acceleration (GA) of animals is defined as the variation in the rate of body weight growth over time (in days), and is given by the second derivative of the animals’ body weight growth function described by the Brody function (or first derivative of AGR) as being equal to $-4343.4\left(0.9242\right){\left(0.0023\right)}^{2}{\mathrm{and}}^{\;\left(-0.0023x_{i}\right)}$ and $-363.13\left(0.9123\right){\left(0.0028\right)}^{2}{\mathrm{and}}^{\;\left(-0.0028x_{i}\right)}$ for males and females, respectively. The graphic representation of GA for males and females is given in Figure 4.

Males showed greater growth acceleration compared to females at the beginning of the growth trajectory, being equal to females at 397 days of age (AC = −0.000852 kg/day), with females showing greater growth acceleration compared to males after that age.

## 4. Conclusions

In view of the above, it is concluded that the nonlinear regression model that uses the Brody function proved to be adequate to represent the body growth curve of Nelore cattle. Similarly, the approach of a nonlinear regression model in the context of mixed models allows for a reduction of the residual variance and an increase in the model’s precision. Thus, males showed higher body weights and greater body mass (BW) gains compared to females, with advancing age. On the other hand, females showed higher TCR compared to males at the beginning of the growth trajectory with opposite behavior after 60 days of age. In addition, males showed greater growth acceleration than females from the beginning of the growth trajectory until 397 days of age, with females showing greater growth acceleration than males after this age.

## Figures and Tables

**Figure 1 animals-13-00101-f001:**
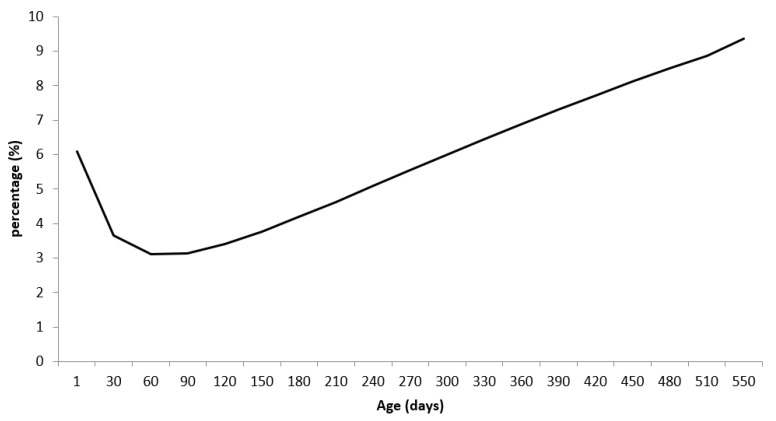
Differences in body weights (in %) between males and females of the Nelore breed, obtained through growth curves by using the Brody function in nonlinear mixed models.

**Figure 2 animals-13-00101-f002:**
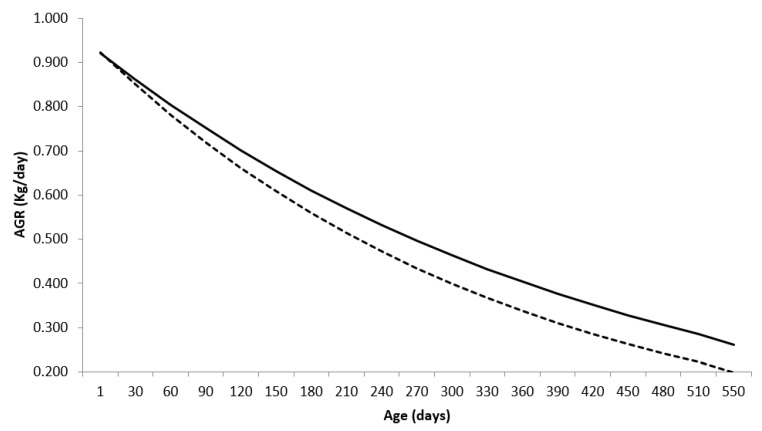
Absolute growth rate (AGR) for Nelore breed males (**—**) and females (**---**), using the Brody function in a mixed nonlinear model.

**Figure 3 animals-13-00101-f003:**
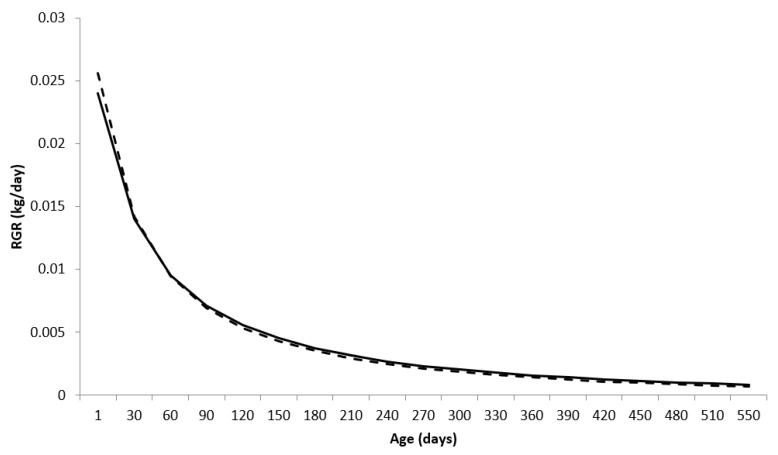
Relative growth rate (RGR) for males (**—**) and females (**---**) of the Nelore breed, using the Brody function in a mixed nonlinear model.

**Figure 4 animals-13-00101-f004:**
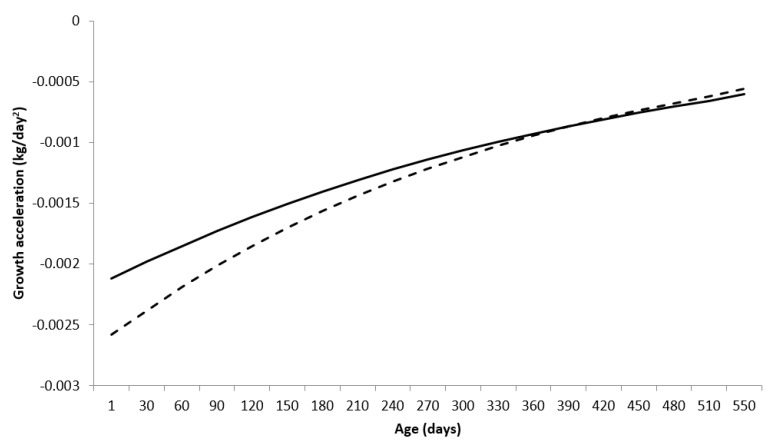
Growth acceleration (GA) for Nelore breed males (**—**) and females (**---**), using the Brody function in a mixed nonlinear model.

**Table 1 animals-13-00101-t001:** Nonlinear regression models used to describe growth curves.

Model	Equation
Brody	Y_t_ = A (1 − Be^−kt^) + Ɛ
Gompertz	Y_t_ = ${\mathrm{Ae}}^{-e^{-B\left(\mathrm{kt}\right)}}$+ Ɛ
Logistic	Y_t_ = A (1 + Be^−kt^)^−1^ + Ɛ
Richards	Y_t_ = A (1 − Be^−kt^)^m^ + Ɛ
Von Bertalanffy	Y_t_ = A (1 − Be^−kt^)^3^ + Ɛ
Von Bertalanffy modified	Y_t_ = A (1 − Be^−kt^)^2^ + Ɛ
Meloun 1	Y_t_ = A − Be^−kt^+ Ɛ

Note: Yt body weight at age t; A asymptotic weight or average weight at maturity; B constant of integration; k maturity rate and m constant that shapes the curve.

**Table 2 animals-13-00101-t002:** Estimates of means, standard deviations (SD), and coefficients of variation (CV) of body weight at birth (WB), 120 (W120), 210 (W210), 365 (W365), 450 (W450), and 550 (W550) days of age, in kg, by gender (2088 males and 3348 females).

Gender	Age	Average (Kg)	SD (Kg)	CV (%)
Male	WB	32.72	3.19	9.76
	W120	131.52	16.75	12.74
	W210	191.79	22.11	11.53
	W365	241.08	27.07	11.23
	W450	279.57	32.24	11.53
	W550	332.63	41.15	12.37
Female	WB	31.70	3.26	10.28
	W120	127.53	15.35	12.03
	W210	184.48	20.85	11.30
	W365	227.13	27.99	12.32
	W450	257.84	29.39	11.40
	W550	301.76	32.63	10.81

**Table 3 animals-13-00101-t003:** Estimates of asymptotic weight (A), in kg, constant of integration (B) and rate of maturation (K), in days, for all nonlinear functions and in each gender.

Model	Gender	A	B	K
Brody	Male	479.45	0.9216	0.0019
Gompertz		357.51	2.0002	0.0051
Logistic		334.00	4.6810	0.0078
Meloun 1		479.45	441.08	0.0019
Von Bertalanffy		376.06	0.5090	0.0040
Von Bertalanffy modified		390.03	0.6695	0.0035
Brody	Female	377.08	0.9067	0.0025
Gompertz		306.93	1.9472	0.0059
Logistic		289.00	4.5291	0.0091
Meloun 1		377.10	341.90	0.0025
Von Bertalanffy		319.30	0.4969	0.0048
Von Bertalanffy modified		328.08	0.6548	0.0042

Note: A, asymptotic weight, in kg; B, constant of integration; K, rate of maturation.

**Table 4 animals-13-00101-t004:** Average square error estimates (ASE), asymptotic standard deviation (ASD), average deviation of residual deviations (ADR), adjusted coefficient of determination (R²aj), asymptotic index (AI), minus two times the natural logarithm of the likelihood function 2log(L) Akaike information criterion (AIC), and Bayesian information criterion (BIC) for each gender and each function.

Functions	Gender	EAS	ASD	ADR	R²aj	AI	−2log(L)	AIC	BIC
	Male								
Brody		808.11	28.43	21.37	0.92	48.87	165.178	165.186	165.217
Gompertz		930.15	30.50	24.28	0.91	53.87	167.615	167.623	167.654
Logistic		1066.04	32.65	26.93	0.90	58.67	169.977	169.985	170.017
Meloun 1		808.11	28.43	21.37	0.92	48.87	165.178	165.186	165.217
V. B. ^1^		884.94	29.75	23.26	0.92	52.09	166.751	166.759	166.790
V. B. M. ^2^		863.51	29.39	22.74	0.92	51.21	166.326	166.334	166.365
	Female								
Brody		644.05	25.38	19.06	0.92	43.51	186.929	186.937	186.969
Gompertz		740.39	27.21	21.41	0.91	47.71	189.729	189.737	189.769
Logistic		855.74	29.25	23.76	0.90	52.12	192.638	192.646	192.677
Meloun 1		644.05	25.38	19.06	0.92	43.51	186.929	186.937	186.969
V. B. ^1^		703.73	26.53	20.53	0.92	46.14	188.709	188.717	188.749
V.B. M. ^2^		686.71	26.21	20.10	0.92	45.39	188.217	188.225	188.257

Note: ^1^ Van Bertalanfy; ^2^ Van Bertalanfy modified. EAS, error average square; ASD, asymptotic standard deviation; ADR, average deviation of residual deviations; R²aj, adjusted coefficient of determination; AI, asymptotic index; 2log(L) minus two times the natural logarithm of the likelihood function; AIC, Akaike information criterion; BIC, Bayesian information criterion for each gender and in each role. Adapted from Giese [16].

**Table 5 animals-13-00101-t005:** Estimates of asymptotic weight (A), constant of integration (B), maturation rate (K), estimates of residual variance components ($\sigma_{e}^{2}$ ), of the asymptotic weight ($\sigma_{a}^{2}$ ), maturation rate ($\sigma_{k}^{2}$) and covariance between asymptotic weight and maturation rate ($\sigma_{a,k}$), logarithm of the likelihood function (−2 Log (L)), Akaike information criterion (AIC) and Schwarz Bayesian information (BIC) for models that used the Brody function in context of mixed and fixed models and by gender.

	Male				Female			
	Mixed		Fix		Mixed		Fix	
Parameter	Value	SE	Value	SE	Value	SE	Value	SE
A	434.40	0.0033	479.45	0.0001	362.13	1.2951	377.08	2.1670
B	0.9242	0.0003	0.9216	0.0009	0.9123	0.0004	0.9067	0.0010
K	0.0023	0.0001	0.0019	0.0001	0.0028	0.0002	0.0025	0.0003
$\sigma_{e}^{2}$	5.8662		28.4300		4.2804		25.3763	
$\sigma_{a}^{2}$	27.8534				28.3107			
$\sigma_{k}^{2}$	0.0003				0.0003			
$\sigma_{a,k}$	0.0045				0.0043			
−2 Log(L)	155.530		165.178		175.488		186.929	
AIC	155.544		165.186		175.502		186.937	
BIC	155.598		165.217		175.558		186.969	

Note: SE: standard error. A, asymptotic standard deviation; B, constant of integration; K, maturation rate; $\sigma_{e}^{2}$, estimates of residual variance components; $\sigma_{a}^{2}$, of the asymtotic weight; $\sigma_{k}^{2}$, maturation rate; $\sigma_{a,k}$, covariance between asymptotic weight and maturation rate, (−2Log (L)), logarithm of the likelihood function; AIC, Akaike’s information criterion; BIC, Schwarz’s Bayesian information criterion.

## Data Availability

The raw data is available upon request.

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
