# Peer review of "Mixed Models in Nonlinear Regression for Description of the Growth of Nelore Cattle"

_animals, 2022, doi:10.3390/ani13010101_

Round 1

Reviewer 1 Report

It is a well-presented research project that compares various non-linear mixed models to better understand the growth of Nelore cattle. Minor changes are needed to the Materials and Methods. There is a font issue in 2.3. Quality of adjustment evaluators of nonlinear models that needs to be addressed.

Additionally, in 2.3. Quality of adjustment evaluators of nonlinear models,  and 2.4. Statistical analysis, the presented models need to be explained in detail.

Author Response

  1. Minor changes are needed to the Materials and Methods. There is a font issue in 2.3. Quality of adjustment evaluators of nonlinear models that needs to be addressed.
    1. As requested by the reviewer, the fonts have been corrected.
  2. Additionally, in 2.3. Quality of adjustment evaluators of nonlinear models,  and 2.4. Statistical analysis, the presented models need to be explained in detail.
    1. The explanation of the models was rewritten to meet what was requested by the reviewer. However, the descriptions of the model components could not be changed.

Reviewer 2 Report

In animal production and genetic selection, it is useful to assess whether in relation to the final goal, the growth is satisfactory for that particular breed and breeding method. Each breed, sex and breeding method has its own peculiarities and predicting the results of breeding can help to correct in time any problems that may affect the qualitative-quantitative characteristics of the product and consequently have considerable economic consequences. The authors applied the proposed method over time to Brazilian production of different breeds and regions in order to provide more precise equations tailored to different production realities.

In my opinion, the method is correct.

The conclusions are consistent with the evidence and arguments presented and they address the main question posed.

References are appropriate because they refer to a particular production reality such as Brazil.

Actually, the work is very specific since it is dedicated to Brazilian production, yet we are talking about the world's second largest producer of beef. Giving space to this work is part of the Animals’ editorial choices. From a scientific point of view, the work is correct and provides interesting results.

The paper needs a minor revision as indicated below.

- nelore or Nellore or Nelore? Please use the same word along the text. I suggest Nelore.

- English needs to be revised.

- Verify reference style (authors, abbreviations,

Line 70. Yt modify in Yt

Lines 88, 92, 93. yij modify in yij

Line 89. tij modify in tij

Line 99.           yiJi modify in yij or explain what is Ji

                        delete = before t’i

                        :::; modify in … ;

                        tiJi modify in tij or explain what is Ji

                        instead of ; use , (yi1, yi2, …, yin)

Line 101. yi/ti modify in yi/ti

Lines 113 – 114. The verb is missing.

Lines 159, 169. R2ij modify in R2ij

Lines 186, 194. Logistic not Logístico

Lines 243-246. The verb is missing.

Line 273. Modify xi in xi

Line 298. Kg/day2 please correct.

Line 303. Modify similarly in Similarly

Author Response

  1. Nelore or Nellore or Nelore? Please use the same word along the text. I suggest Nelore. 
    1. Due to the reviewer's request, all nomenclatures were changed to Nelore.
  2. English needs to be revised.
    1. In compliance with what was requested by the reviewer, the text was fully reviewed by a trained professional hired for this purpose.
  3. Verify reference style (authors, abbreviations)

    1. Following the reviewer's recommendation, all references were reviewed and standardized in accordance with the journal's standards.
  4. Line 70. Yt modify in Yt

    1. Modified as recommended by the reviewer.
  5. Lines 88, 92, 93. yij modify in yij

    1. Modified as recommended by the reviewer.
  6. Line 99.           yiJi modify in yij or explain what is Ji

                            delete = before t’i

                            :::; modify in … ;

                            tiJi modify in tij or explain what is Ji

                            instead of ; use , (yi1, yi2, …, yin)

    1. Modified as recommended by the reviewer.
  7. Line 101. yi/ti modify in yi/ti

    1. Modified as recommended by the reviewer.
  8. Line 101. yi/ti modify in yi/ti

    1. Modified as recommended by the reviewer.
  9. Lines 113 – 114. The verb is missing.

    1. Modified as recommended by the reviewer.
  10. Lines 159, 169. R2ij modify in R2ij

    1. Modified as recommended by the reviewer.
  11. Lines 186, 194. Logistic not Logístico

    1. Modified as recommended by the reviewer.
  12. Lines 243-246. The verb is missing.

    1. Modified as recommended by the reviewer.
  13. Line 273. Modify xi in xi

    1. Modified as recommended by the reviewer.
  14. Line 298. Kg/day2 please correct.

    1. Modified as recommended by the reviewer.
  15. Line 303. Modify similarly in Similarly

    1. Modified as recommended by the reviewer.
